# Comparative Investigation of Conventional and Innovative Headspace Extraction Methods to Explore the Volatile Content of Human Milk

**DOI:** 10.3390/molecules27165299

**Published:** 2022-08-20

**Authors:** Sarah Le Roy, Catherine Fillonneau, Benoist Schaal, Carole Prost, Angélique Villière

**Affiliations:** 1Oniris, Nantes Université, CNRS, GEPEA, UMR 6144, Flavor Plateform, F-44322 Nantes, France; 2Centre for Smell, Taste and Feeding Behaviour Science, Developmental Ethology & Cognitive Psychology Laboratory, CNRS (UMR 6265), Université de Bourgogne-Inra-ASD, F-21000 Dijon, France

**Keywords:** headspace extraction, solid phase micro-extraction, dynamic headspace analysis, human milk, volatilome, olfaction, neonate

## Abstract

The odor of human milk induces search-like movements and oral activation in newborns, which increases their chances of taking advantage of milk intake and benefits. However, the underlying volatile fraction of human milk remains understudied. This study aimed to devise a simple method to extract a wide range of volatile compounds from small-volume human milk samples. Headspace solid phase micro-extraction (HS-SPME) with a Car/PDMS fiber and dynamic headspace extraction (D-HS) with a Tenax or a trilayer sorbent were tested because of their selective affinity for volatiles. Then, innovative variations of these methods were developed to combine their respective advantages in a one-step extraction: Static headspace with multiple SPME fibers (S-HS-MultiSPME), Dynamic headspace with multiple SPME fibers (D-HS-MultiSPME) and dynamic headspace with multiple SPME fibers and Tenax (D-HS-MultiSPME/Tenax). The extracts were analyzed by gas chromatography coupled with mass spectrometric and flame ionization detection. The relative performances of these methods were compared based on qualitative and semi-quantitative analyses of the chromatograms. The D-HS technique showed good sensitivity for most compounds, whereas HS-SPME favored the extraction of acids. The D-HS-MultiSPME/Tenax identified more than 60 compounds from human milk (some for the first time) and evidence of individual singularities. This method that can be applied to volatilome analysis of any biological fluid should further our understanding of human milk odor.

## 1. Introduction

The possibility for human newborns to get the benefits of human milk depends on the successful establishment of the interactions leading to breastfeeding [1]. In this regard, the odor of human milk was shown to induce active searching, rooting, mouthing, and sucking behavior in newborn infants [2,3,4], but also to elicit calming effects which facilitate the coordinated behavior that ends in successful latching and milk intake [5]. In the longer term, human milk odor was also shown to influence the development of subsequent food preferences and choices in weanling infants [6,7]. Despite this potentially important communicative role of conspecific milk odor, the volatile fraction that supports it remains poorly studied [3]. Compared to the hundreds of compounds reported in cow’s milk or derived formulas by hundreds of papers [8,9,10], much fewer volatiles have been identified in human milk [11], suggesting a qualitatively and/or quantitatively contrasted volatilome and that a great part of the human milk volatilome remains underexplored. Thus far, the volatiles of human milk were mostly traced in the context of environmental contamination of the first food [12,13,14,15], including volatile organic compounds (VOCs; e.g., chloroform, benzene and toluene) or semi-volatile organic pollutants [POPs; e.g., polychlorinated biphenyls (PCBs), polychlorinated dibenzo-*p*-dioxins (PCDDs) and polychlorinated dibenzofurans (PCDFs)]. Tobacco smoke also leads to the transfer of pollutants into breastmilk, especially nicotine [16,17]. The presence of nitro-musk’s was related to their use as fragrances in cosmetics, soaps or laundry products [18]. Besides, some flavor compounds (e.g., d-carvone, l-menthol or trans-anethole) from the mother’s diet were shown to pass into milk [19]. Finally, the profile of VOCs in ejected breast milk is easily modified by heating processes and storage conditions, leading to a Maillard reaction and lipid oxidation [20,21,22,23,24]. Among VOCs, carboxylic acids were estimated to represent up to 80% of the total VOC content of human milk [25]. The presence of aldehydes, ketones, alcohols, furans, lactones, nitrogen-containing compounds, esters and terpenes reveals the considerable chemodiversity of human milk effluvium [11,25,26,27,28]. Most of these volatiles occur in low, even traces concentrations. Subsequent difficulties of chemical identification are increased by the generally rather small volumes available of fresh human milk samples. Thus, before making assumptions about their behavioral activity in neonates, VOCs’ composition in human milk needs more investigation.

Different extraction methods were previously used to trap volatiles from human milk before GC analysis, such as solvent extraction [29], solvent-assisted flavor evaporation (SAFE) [23,24,30,31,32], or simultaneous distillation-extraction (SDE) [25]. These methods led to identifying compounds from the main chemical classes in human milk, but the elimination of the solvent and the temperatures generally applied (>50 °C) may have induced a loss of VOCs [33]. In addition, the amount of sample required was often large, reaching up to one liter [25]. Hence, milk from multiple mothers was often pooled before extraction, limiting the characterization of individual samples’ chemodiversity and excluding the characterization of between-mother variations. 

Recently, headspace extraction techniques have been favored to analyze the volatile fraction of human milk, making solvents unnecessary and extraction temperatures and sampling volumes much lower. Among these methods, headspace solid phase microextraction (HS-SPME) exploited the high ad/bsorption power of a fused silica fiber coated with a specific extraction sorbent material [34]. This method allows several combinations of sorbents onto fibers to extend the range of chemical classes extracted for several biological matrices [35]. The divinylbenzene/Carboxen/polydimethylsiloxane (DVB/Car/ PDMS) fiber allowed the extraction of various compounds from processed human milk [20,21]. Still, the car/PDMS fiber recovered the largest profile of VOCs from dairy products [36,37]. Moreover, this type of fiber allowed extracting a profile of odorants resembling the one of the original matrix [38], suggesting that it tends to trap a representative volatile profile. However, the small amount of coating material on SPME fibers relates to limited sensitivity [35]. Compounds can be pre-concentrated in a prior step of extraction to increase the extraction capacity of the HS-SPME. For example, solid phase extraction (SPE) was applied to pre-concentrate target compounds from human milk before using HS-SPME extraction [39]. However, due to polarity compatibility issues of the two consecutive extractive materials, a loss of some VOCs may have occurred, limiting the use of this method in a non-targeted approach. The stir-bar sorptive extraction (SBSE) and its headspace form, the headspace sorptive extraction (HSSE), were also used to resolve the lack of sensitivity of SPME. Being coated with a greater volume of the extraction phase, the stir-bar allows the recovery of more volatiles and decreases to 5 mL the volume of human milk required for analysis [26]. However, the dual-phase equivalence of Car/PDMS SPME fibers (i.e., polar and apolar) for the SBSE bar lacks availability, and the apolar PDMS phase remains the polymer currently used in SBSE methods, making it difficult to implement for recovering analytes that are both polar and highly volatile [40].

Dynamic headspace (D-HS) extraction is a valuable approach for the non-targeted exploration of analytes from various matrices because of the wide range of available sorbents and the large amounts of extractive materials. The sample is purged with an inert gas to sweep out volatiles and trap them on a sorbent. Accordingly, the sample’s headspace is constantly renewed, leading to the volatile enrichment of the trap and correlative depletion in the sample. Tenax was used for many applications because of its low selectivity and thermal stability [41]. It enabled the identification of several compounds in human milk [27,28]. However, due to its molecular size screening and hydrophobicity, Tenax is limited for extracting highly volatile and polar molecules. Hence, tubes packed with a series of discrete beds of different sorbent materials with increasing strength, such as graphitized carbon black or carbon molecular sieves, were developed to increase the range of extracted compounds [41]. However, such multibed traps were suboptimal for extracting volatiles from cow’s milk since much vapor is retained on these traps leading to water issues in the desorption system [42]. Besides, the combination of Tenax and Carbograph 5TD, which is supposed to trap smaller compounds (C < 7), does not lead to the identification of additional compounds in human milk compared to the use of Tenax alone [43]. Another strategy was applied to obtain a more exhaustive volatile profile of human milk, using different methods with complementary extractive properties on several sample replicates. For example, the analysis of SBSE and HSSE extracts from human milk allowed to cover compounds having a wide range of boiling temperatures [26]. Similarly, dynamic and static methods were both used to extract the low and medium/high boiling temperature compounds, respectively, and extensively study the volatile fraction of cheeses [44]. Such strategies require several steps of extraction, multiply the number of chromatograms to be exploited, and increase the handling and processing times as well as the sample volume required. In sum, the approaches used so far to analyze the volatilome of human milk bear disadvantages (such as the polarity- or size-screening of compounds, the large volume of samples needed or the multiple steps of extraction) which limit their use in an effective non-targeted exploration of low-volume samples of human milk.

In this context, the present study aimed to devise a method to extract a wide range of volatile compounds from small volumes of human milk. Bearing in mind the behavioral function of milk odor, headspace methods were considered as they allow extraction of sufficiently volatile compounds to activate neonatal olfaction. Ideally, to limit sample loss and with a perspective of automation, this method should work in only one step of extraction. Two conventional headspace extraction methods, HS-SPME and D-HS, were evaluated in the analyses of human milk. HS-SPME was tested with a Car/PDMS fiber, while the D-HS method was evaluated with the Tenax sorbent (D-HS-Tenax) and compared with a trilayer sorbent (D-HS-trilayer). Several innovative variations of these methods were developed to combine their respective advantages in only one step of extraction. Static headspace with multiple solid-phase microextraction fibers (S-HS-MultiSPME), dynamic headspace with multiple solid-phase microextraction fibers (D-HS-MultiSPME) and dynamic headspace with multiple solid-phase microextraction fibers and Tenax (D-HS-MultiSPME/Tenax). These six methods were compared based on the VOCs profiles obtained for human milk.

## 2. Results and Discussion

### 2.1. Comparison of Headspace Extraction Methods

The GC profiles of extracts obtained in static or dynamic headspace using conventional HS-SPME-Car/PDMS (static), D-HS-Tenax (dynamic), and D-HS-trilayer (dynamic) were first analyzed and compared. Then, three variations of the latter methods were applied, including (1) a static headspace extraction with ten Car/PDMS SPME fibers placed simultaneously into the headspace of the milk sample (S-HS-MultiSPME); (2) a dynamic headspace extraction using the same number and type of SPME fibers (D-HS-MultiSPME); and (3) a dynamic extraction procedure using a tube containing Tenax sorbent enhanced with these 10 SPME fibers (D-HS-MultiSPME/Tenax). The VOCs profiles of extracts obtained by these three innovative headspace extraction methods were also analyzed and compared from a qualitative and semi-quantitative point of view. Following a non-targeted approach aiming to detect a maximum of compounds in breast milk, the mass spectrometer was used in TIC (total ionic current) mode to obtain qualitative information on the identity of the detected compounds. However, data from the flame ionization detector (FID) were preferred for a semi-quantitative approach, in accordance with good practices [45,46] and standard recommendations [47]. Indeed, although the response factors are different from one molecule to another for FID, the disparities observed in the response factors for different compounds are smaller than for the mass spectrometer and generate less distortion of the chromatographic profiles with respect to the real compositional profile of the extract or product analyzed.

The resulting profiles of the six methods reveal that the recovery of volatile compounds and their chemical classes differ significantly according to the extraction method, as shown by the FID area and mass spectrometry responses (Table 1). Summarizing these results, Figure 1 represents the profiles of the volatile compounds (total peak area per chemical class) for each extraction method (except for the D-HS-trilayer method that leads to unexploitable results; see explanation hereafter). In total, 54 compounds were identified from at least eight chemical classes.

#### 2.1.1. Static Headspace Solid Phase Micro Extraction with Car/PDMS Fiber (HS-SPME-Car/PDMS)

The number of compounds identified (*n* = 17) and the total FID peak area (TPA) associated with the volatile profile obtained by this method is relatively weak (Figure 1). Aldehydes, carboxylic acids and ketones are the major families of compounds extracted by this method and represent more than 85% of the TPA. Pentanal (linear retention index, LRI: 980), hexanal (LRI: 1088) and benzaldehyde (LRI: 1563) represent 38% of the total chromatographic peak area associated with the HS-SPME extract (Table 1). In human milk, most aldehydes are oxidation products of oleic, linoleic and linolenic acids [25]. The following free fatty acids (acetic (LRI: 1460), butanoic (LRI: 1637), hexanoic (LRI: 1862), octanoic (LRI: 2076) and decanoic (LRI: 2172) acids) are also identified in the extract, representing more than 30% of the TPA. Among them, octanoic and decanoic acids are the most abundant in the present milk sample, in line with previous results on human milk analyzed with DVB/Car/PDMS SPME fibers [21]. Carboxylic acids are released from triglycerides and phospholipids by the action of lipases [25]. Besides, 2-propanone (LRI: 793), 2-butanone (LRI: 904) and 2,3-butanedione (LRI: 975) were also identified in the extract. These compounds represent 18% of the TPA associated with this HS-SPME extract and are considered to be degradation products from unsaturated fatty acid oxidation in dairy products [48]. Compounds from other chemical classes are detected in the volatile profile obtained with this milk extraction procedure, but to a lesser extent. Limonene (LRI: 1210) which is regularly detected in SPME extracts of human milk [39,40], and 1,8-cineole (LRI: 1220) were also detected (<5% of TPA). Terpenes in breast milk are presumed to derive from environmental exposure (diet, cosmetics, skin care products) [27]. Furfural (LRI: 1486) is also detected in this extract (5% of the TPA). Already detected in human milk using other methods than HS-SPME extraction [19,20,21,28], this compound is usually strongly related to heating as a Maillard product [27,49]. Furthermore, γ-butyrolactone (LRI: 1685) is also identified in this HS-SPME extract. Several lactones have already been detected in human milk after SPME extraction but not γ-butyrolactone [20,21], which was only detected in dairy products [50]. The origin of lactones in human milk is not elucidated, but the literature on cow’s milk suggests these compounds could derive from fat [48]. The only alcohol detected in the HS-SPME-Car/PDMS extract is 1-pentanol (LRI: 1250, <1% of the TPA). In human milk, most alcohols come from the degradation of unsaturated fatty acids [25]. Styrene (LRI: 1275), also detected in the extract, was previously reported in human body fluids as an environmental contaminant [12,13,28]. Additionally, despite the great care taken during analyses (pre-cleaning glass vial, using hermetic screw-caps), some detected styrenes might originate in the laboratory environment or milk expression materials. However, this compound is regularly reported in breast milk even after corrections accounting for possible contamination format different stages (milk sampling, storage, or analysis) [12].

#### 2.1.2. Dynamic Headspace with Tenax (D-HS-Tenax)

The visualization of the D-HS-Tenax extract profile reveals a strong increase in the TPA, which is enhanced by a 7-fold ratio compared to the HS-SPME-Car/PDMS extract. Moreover, 32 compounds were additionally detected in this D-HS-Tenax extract. Thus, this method can strongly increase the qualitative and quantitative recovery of compounds from human milk, in line with previous literature comparing these methods in dairy products [51]. Furthermore, the distribution of chemical families is significantly modified relative to the precedent extract, particularly for terpenes, alcohols, aldehydes and carboxylic acids. Indeed, terpenes represent 3% of the TPA in the SPME extract but represent more than 20% in the present D-HS-Tenax extract. Six terpenes [β-pinene (LRI: 1114), 3-carene (LRI: 1156), β-myrcene (LRI: 1163), β-phellandrene (LRI: 1174), γ-terpinene (LRI: 1259), and α-terpinolene (LRI: 1295)] are additionally detected in the D-HS-Tenax extract. Moreover, limonene and 1,8-cineole, identified in both extracts, are 40 times higher in the D-HS-Tenax extract. The increase in recoveries of terpenes, which are mainly apolar, may be explained by the larger amount of the hydrophobic sorbent (Tenax-200 mg) used in the D-HS method than in the SPME fiber (PDMS < 0.5 µg). Furthermore, the dynamic mode of extraction could also be involved in this difference. Alcohols represent less than 1% of the TPA of the SPME extract but represent more than 10% of the volatile profile obtained with the present D-HS extraction. Five alcohols are additionally detected in the D-HS extract: 2-methyl-1-propanol (LRI: 1077), 1-butanol (LRI: 1140), (*E*)-2-penten-3-ol (LRI: 1154), 3-methyl-1-butanol (LRI: 1205) and 2-ethyl-1-hexanol (LRI: 1488). Moreover, 1-pentanol, detected in D-HS and SPME extracts, is found at a significantly higher level in the D-HS-Tenax extract with a 23-fold ratio increase. Aldehydes remain the most represented chemical class in both extracts. Their proportion increases from 38% for SPME to 56% in the present D-HS extract. Ten additional aldehydes were identified compared to the previous extract: heptanal (LRI: 1190), (*E*)-2-hexenal (LRI: 1233), octanal (LRI: 1295), (*E*)-2-heptenal (LRI: 1346), nonanal (LRI: 1409), (*E*)-2-octenal (LRI: 1445), (*E*,*E*)-2,4-heptadienal (LRI: 1484), decanal (LRI: 1498), (*E*,*Z*)-2,4-heptadienal (LRI: 1521), (*E*)-2-nonenal (LRI: 1563). Moreover, most aldehydes detected in both extracts were found significantly higher in the D-HS-Tenax extracts with a 10 mean ratio. The better sensitivity of D-HS for terpenes, alcohols and aldehydes than HS-SPME was previously also noted in cola beverages or dairy products [44,52,53]. In addition, esters [ethyl acetate (LRI: 878), methyl decanoate (LRI: 1601) and ethyl decanoate (LRI: 1644)] were only detected in the profile obtained with the D-HS-Tenax extraction. However, they represent less than 1% of the total volatile profile. These compounds originate from the esterification of fatty acids [54]. Pyrazine (LRI: 1226) is also identified in the profile of the D-HS-Tenax extract. Pyrazines had never been instrumentally detected in human milk. However, an olfactometric analysis found one methoxypyrazine in human milk that can be more sensitive than instrumental detection for compounds presenting a low odor threshold. Pyrazines are found in numerous products, especially roasted food products such as coffee [8]. Therefore, detecting these compounds in a human milk sample could be linked to maternal ingestion [11]. 

Conversely, carboxylic acids, which represented more than 30% of the TPA of the SPME extract, drop to 1% of the volatile profile obtained with the present D-HS-Tenax extraction. Two carboxylic acids were not even detected in the D-HS-Tenax extract. Thus, the HS-SPME extract with a Car/PDMS fiber is more efficient for extracting free fatty acids than D-HS-Tenax. Similar observations were made when DVB/Car/PDMS SPME fiber and D-HS-Tenax were compared for the volatile fraction extraction of dairy products [44]. Indeed, the polar Carboxen sorbent makes these SPME fibers more efficient for extracting polar molecules such as short-chain fatty acids compared to the apolar Tenax. This limitation of Tenax for extracting polar compounds has previously been underlined [41]. Finally, the 6 following ketones were detected in the D-HS extract: 3-octanone (LRI: 1264), 6-methyl-5-hepten-2-one (LRI: 1351), 2-nonanone (LRI: 1397), 3-octen-2-one (LRI: 1423), 3, 5-octadien-2-one (LRI: 1525), 2-undecanone (LRI: 1611). However, the 2-propanone, which was strongly represented in the HS-SPME extract, is not detected in the present D-HS extract.

As an intermediary outcome, the D-HS-Tenax extraction enables more compounds to be detected and, to a greater extent, than the HS-SPME extraction with a Car/PDMS fiber. This result can be partly explained by the greater quantity of sorbent used in the D-HS cartridge (200 mg) compared to an SPME fiber (0.5 µg). However, the D-HS-Tenax method displays a lack of short-chain acid recovery. This chemical family is characteristic of milk. Some of its representatives, namely butanoic and nonanoic acids, were previously detected by newborn infants when presented as pure compounds [55,56]. Moreover, newborns appear to be more sensitive than adults to short-chain carboxylic acids found in sweat [57]. Hence, carboxylic acids are compounds not to overlook to choose an extraction method suitable for analyzing human milk’s volatile fraction. Therefore, the interest in D-HS-Tenax is mitigated regarding the objective of a non-targeted exploration of the volatile fraction of human milk. Conversely, the presence of the polar Carboxen in Car/PDMS SPME fibers enables short-chain acids to be recovered.

#### 2.1.3. Dynamic Headspace Extraction with a Trilayer Cartridge Combining Tenax TA, Carbograph, and Carboxen (D-HS-Trilayer)

To extend the range of compounds extracted by the dynamic headspace procedure to more polar compounds, a custom-assembled cartridge built in associating Tenax and the more polar Carboxen 1003 in powder form was evaluated. The composite cartridge thus obtained was completed by the third layer of Carbograph 1TD sorbent, ensuring the extraction of compounds from the range C5–C7 covered neither by Carboxen 1003 (C2–C5) nor by Tenax (C7–C30). Hence, this trilayer cartridge was tested in a dynamic extraction mode according to the extraction and desorption conditions established for the D-HS-Tenax extraction. 

Despite optimization of the extraction procedure aiming to limit water trapping, the highly polar Carboxen in the trilayer tube have excessively strong water vapor retention leading to the ice-plug formation in the cryotrap desorption system leading, thus to unreproducible chromatograms. Such water issues were reported in a previous study using combinations of carbon molecular sieves to extract volatile compounds from cow’s milk [42]. Hence, the association of apolar Tenax with the commercial Carboxen in a powdered form in the D-HS-trilayer method does not lead to exploitable results. 

Therefore, three innovative variations of HS-SPME and D-HS methods were designed to combine the high concentrative power of the D-HS-Tenax method and the large polar range recovery of the HS-SPME-Car/PDMS.

#### 2.1.4. Static Headspace Using Multiple Solid Phase Microextraction Car/PDMS Fibers (S-HS-MultiSPME)

HS-SPME with a Car/PDMS fiber allows the extraction of a large range of compounds including carboxylic acids but lacks sensitivity for most of the volatile compounds in human milk. Thus, considering the weak quantity of sorbent used in the SPME procedure and the fact that the more the thickness of the SPME fiber is increased, the more the sensitivity is improved [34], a way to increase the quantity of sorbent was sought. However, Car/PDMS fibers with higher capacity are not commercially available due to mechanical issues and the thermal stability of the sorbent film [58]. Headspace sorptive extraction (HSSE) methods were not envisaged, despite the presence of a higher quantity of sorbent. Indeed, stable polar sorbents are not commercially available, restricting the concentrative effect to apolar compounds. Hence, in this study, we used a greater number of Car/PDMS fibers in their original structure to increase the sensitivity of the method toward polar compounds. Thus, an extraction procedure in a static mode using 10 Car/PDMS SPME fibers was evaluated and referred to as the S-HS-MultiSPME method. 

The volatile profile of compounds obtained using 10 Car/PDMS SPME fibers in a static extraction procedure is characterized by a 6-fold increase of the TPA compared to the use of a single fiber. The peak areas obtained for some compounds with this procedure are close to those obtained with the D-HS-Tenax method due to the increase in the quantity of sorbent material. This increase in the sensitivity of S-HS-MultiSPME is significant for all chemical families except for O-heterocycles and ketones. A larger number of aldehydes, ketones, alcohols, N-heterocycles and one ester are additionally identified in this S-HS-MultiSPME extract compared to the extract obtained with the conventional HS-SPME procedure. The highest sensitivity improvement is noted for carboxylic acids. Four additional carboxylic acids [propanoic (LRI: 1547), pentanoic (LRI: 1746), heptanoic (LRI: 1968) and nonanoic acids (LRI:2172)] are identified when 10 SPME fibers are used instead of one. The proportion of these compounds reaches up to 62% of the TPA. 

For most of these chemical families of compounds, the recovery is increased from 2 to 10 times when the number of fibers is increased from 1 to 10. The lowest sensitivity increase is noted for terpenes, but no additional terpene was detected with increased SPME fiber quantity. In sum, using 10 SPME fibers (instead of one) in a static extraction mode enables a strong amplification of the signal recovered without deeply altering the relative distribution of chemical classes.

#### 2.1.5. Dynamic Headspace Using Multiple Solid Phase Microextraction Car/PDMS Fibers (D-HS-MultiSPME)

SPME extraction was shown to be potentially more sensitive when used under dynamic conditions through a non-equilibrium sampling process [59]. Thus, a second variation was implemented with 10 SPME fibers used under dynamic conditions. 

The volatile profile obtained for the present D-HS-MultiSPME extraction is characterized by a 4-fold increase of the TPA compared to the HS-SPME using a single Car/PDMS fiber. This gain is lower than the gain observed when the 10 SPME fibers were used under static conditions. However, if using 10 SPME fibers under static conditions leads to better signal amplification, the dynamic conditions significantly modify the proportion of the chemical classes of compounds extracted (Figure 1). For example, carboxylic acids are recovered with a ratio 8-fold higher in the S-HS-Multi-SPME than in the D-HS-MultiSPME method. However, except for carboxylic acids and ketones, the use of SPME fibers in a dynamic configuration leads to a better recovery of the main chemical families of human milk compared to their use in the static configuration. Terpenes are thus recovered to a significantly larger extent under dynamic (21% of the TPA) versus static (<2% of the TPA) conditions. A similar tendency is noted for most alcohols. The same trend also appears to a lesser extent for aldehydes, but three of them (i.e., (*E*)-2-octenal, (*E*,*E*)-2,4-heptadienal and (*E*,*Z*)-2,4-heptadienal) were additionally detected in the dynamic mode. Regardless of the sorbent used, these results highlight the impact of static vs. dynamic extraction conditions for such a matrix. 

Besides, the nature of the sorbent (SPME sorbent vs. Tenax) has a non-negligible impact on the profile of volatiles obtained in dynamic extraction, depending on the chemical classes of compounds. Indeed, when used in a dynamic extraction, the Car-PDMS sorbent from SPME fibers increases the recovery gain of carboxylic acids from less than 2% to 11% compared to Tenax, and six additional acids are detected. Hence, Car-PDMS from SPME fibers are a valuable sorbent to extract carboxylic acids from human milk. 

Thus, an increased number of Car/PDMS fibers leads to a significant increase of the TPAs whatever the static or dynamic extraction mode. Implementing 10 fibers under static conditions leads to a simple signal amplification compared to the conventional HS-SPME with a single fiber. At the same time, their implementation in a dynamic mode significantly changes the proportion of chemical classes of compounds recovered, especially for carboxylic acids, terpenes and alcohols. Hence, the volatile profile acquired through D-HS-MultiSPME gets closer to the profile obtained with D-HS-Tenax. The Car-PDMS SPME fibers allow recovering carboxylic acids to a significantly higher extent than D-HS-Tenax even in dynamic extraction, confirming the impact of the sorbent used in a dynamic extraction on the volatile profile obtained. Finally, these experiments confirm the efficiency of Car/PDMS SPME fibers in extracting carboxylic acids, whatever the mode of extraction applied (static/dynamic). 

#### 2.1.6. Dynamic Headspace Combining Multiple Car/PDMS SPME Fibers and Tenax (D-HS-MultiSPME/Tenax)

Considering the complementarity of Tenax and Car/PDMS from SPME fibers towards the volatile compounds of human milk, the last strategy consisted of combining multiple SPME fibers and Tenax in a single cartridge under a dynamic extraction procedure. 

The volatile profile obtained from this D-HS-MultiSPME/Tenax extraction procedure was associated with the highest TPA compared to all other extraction procedures in the present study. Moreover, this method detects the highest number of compounds (*n* = 53). This performance can be explained by the largest quantity of sorbents used in this method and the complementarity of Car-PDMS and Tenax polarity towards milk volatiles.

This volatile profile is characterized by a large proportion of aldehydes (52%), terpenes (20%), alcohols (10%) and ketones (9%). The relative proportion of these chemical classes is very close to those obtained with the two other dynamic extraction procedures. Although detected to a small extent, esters and N-heterocycles are better detected with the D-HS-MultiSPME/Tenax procedure than with the other methods tested here. Moreover, this method identified one additional compound, the 2-methylpyrrole (LRI: 1575), not previously detected with the D-HS-Tenax or the D-HS-MultiSPME methods. From this chemical class, only pyrrole was previously identified in human milk [25]. This chemical class was largely reported in roasted food products [8] and may be traced to diet. 

Furthermore, this method efficiently recovers carboxylic acids, which were poorly extracted with Tenax only. Therefore, the association of the Car-PDMS SPME fibers with Tenax in the one-step D-HS-MultiSPME/Tenax procedure combines a large range of sorbent polarity and allows the powerful extraction of compounds from a large range of chemical classes, including carboxylic acids. Hence, it results in a more comprehensive profile than those obtained with the methods assessed previously.

### 2.2. Repeatability of the Extraction Methods

Table 1 displays the coefficients of variation (CVs) obtained for the volatile compounds extracted using the above-mentioned extraction methods, except for the D-HS-trilayer method, which leads to non-exploitable results. Carboxylic acids display a higher CV compared to compounds from other chemical classes, whatever the extraction method used. These compounds are strongly retained on polar columns, such as the DB Wax used here, leading to spreading shaped peaks, rendering integration complicated. 

The CVs of compounds extracted by D-HS-Tenax are lower than the CVs obtained for the HS-SPME-Car/PDMS method (CVmean 13% < 35%, respectively). Indeed, the smallness of peaks obtained with the HS-SPME-Car/PDMS method leads to unavoidable variations in the integration of the peak area between replicates leading to high values of CVs. This result contrasts with several papers comparing these two methods on bovine milk [44,51]. However, the volumes of milk used in these studies were higher and did not result in this bias. Regarding the static and dynamic MultiSPME methods, the mean CV of compounds is 18 and 21%, respectively. The repeatability of these methods is improved compared to using a single fiber under static conditions, which the higher associated peak areas can explain. Finally, the mean CV for the D-HS-MultiSPME/Tenax is 15%, which is in the range of the CV obtained from the D-HS-Tenax procedure and in the range of those obtained from validated extraction methods commonly applied to dairy products [44,51,60]. 

In conclusion, among the six extraction procedures tested in the present study, i.e., HS-SPME-Car/PDMS, D-HS-Tenax, D-HS-trilayer, S-HS-MultiSPME, and D-HS-MultiSPME and D-HS-MultiSPME/Tenax methods, the latter leads to the best performance in terms of the range of identified compounds and extraction capacity. The dynamic mode used in the D-HS-MultiSPME/Tenax method enables us to extract a large number of terpenes. In contrast, the use of SPME fibers enables us to recover a large number of carboxylic acids. In addition, the large quantity of Tenax in the trap ensures a high recovery of aldehydes, ketones and alcohols. Finally, this latter method demonstrates suitable repeatability, which is in the range of currently applied methods [44,51,60]. Among the different methods tested, the D-HS-MultiSPME/Tenax is the most efficient way to explore human milk’s volatile fraction. This method was further evaluated from a discriminability point of view.

### 2.3. Discriminative Value of the D-HS-MultiSPME/Tenax Method for Individual Human Milk Samples 

The discriminative potency of the DHS-multi SPME/Tenax method was evaluated based on the volatile profiles obtained from four individual samples of human milk, analyzed in triplicate. The PCA performed on peak areas of the detected compounds (Figure 2) displays the proximity between replicates, confirming the good repeatability of the D-HS-MultiSPME/Tenax method.

In addition, the four individual milk samples are distinctly positioned on the map. Milk samples #2 and #4 are opposed on the first axis, representing 43% of the variance and positively correlated to the content of aldehydes, alcohols, and ketones. These compounds are mostly derived from the oxidation of lipids from milk [25]. Hence, this analysis suggests that milk #4 has begun to undergo lipid oxidation despite the precautions taken to collect the samples. The quantitative and qualitative composition of fatty acids in human milk strongly varies between women, depending on several causes, including genetic, make-up, food intake, stress, and lactational stage [61]. These differences can also be attributed to milk sampling, pooling or freezing conditions before storage. Besides, milk #1 and #3 are opposed on axis 2, representing 19% of the variance. This axis is positively correlated to the presence of terpenes. Thus, the analysis suggests that milk #1 and #3 have the highest and the lowest level of terpenes, respectively. Different diets of mothers could explain these differences in terpene levels [27]. Overall, these observations illustrate the discriminative validity of the D-HS-MultiSPME/Tenax method for small volumes (i.e., 5 mL) of individual human milk samples. 

In addition, the D-HS-MultiSPME/Tenax method enables the detection of 60 compounds from individual human milk samples in only one step of extraction. Representatives of all main chemical families reported in the literature on human milk are recovered by this combinatorial method. Indeed, numerous aldehydes (*n* = 13), carboxylic acids (10), terpenes (10), ketones (9), alcohols (7), and esters (2) were identified. In addition, several minor compounds in human milk, such as furans, lactones, pyrazine and pyrrole compounds, are also recovered and identified through this D-HS-MultiSPME/Tenax method. Moreover, to the best of our knowledge, nine compounds are unanticipated in human milk. Among them, methanoic and propanoic acids, 2-methyl-1-propanol, 3-methyl-1-butanol and γ-butyrolactone belonged to chemical families generally linked to lipid oxidation in milk [25]. β-phellandrene is also detected for the first time in the milk of at least one of the mothers. As a terpene, this compound may derive from a maternal diet [19] or cosmetic usage. Similarly, pyrazine and 2-methylpyrrole are first found in one mother’s milk. Although these chemicals were largely reported in roasted food products [8], their origin in human milk is not clear. Finally, phenol is also reported in this study. Phenols were already detected in HS-SPME extracts of human milk [20,21], but even if they may derive from forage in bovine milk [54], their origin in human milk remains unexplained. 

Hence, the compounds identified in the present study correspond to approximately 30% of the total volatile compounds currently found in human milk [11]. This proportion must be considered in the light of the objective that was mainly centered on volatile and semi-volatile compounds (LRI < 2600). Therefore, heavy compounds like steroids or some pollutants, which are largely reported in studies focusing on specific compounds related to environmental contamination or medication, will not be detected by this method [11,62,63]. Given the high variation in milk volatile profile between women [27] and the restricted number of individual human milk samples analyzed here, the number of compounds identified in human milk is expected to be further increased with the analysis of additional individual milk samples using this innovative D-HS-MultiSPME/Tenax method. Finally, combining this extraction method with chromatographic analysis coupled with a more sensitive or complementary means of detection would probably allow tracing an even greater number of compounds. Time-of-flight mass spectrometry or olfactometry could be considered to detect compounds present in trace amounts but with odorant potential [3].

## 3. Materials and Methods

### 3.1. Chemicals

Chemical standards, including *n*-alkanes (C5–C25), were purchased from Sigma Aldrich (St.-Quentin-Fallavier, France) with purity > 98%. Ultrapure water used in the study was obtained by a Millipore-Q system (Millipore Corp., Saint-Quentin, France).

### 3.2. Samples

Human milk was obtained from healthy volunteers who signed informed consent (*n* = 5). Samples were of mature milk collected on 2 to 12 months postpartum (volume ranged from 20 to 150 mL per woman). The mothers collected the milk into glass pre-cleaned vials [21.2 mL, 23 mm (diameter) × 75.5 mm (height); Chromoptic, Courtaboeuf, France] sealed with a PTFE cap, frozen at −18 °C immediately after collection for storing up to 1-week maximum. When brought to the laboratory, the samples were defrosted 10 min in an ice bath, and aliquots of 5 mL were distributed into pre-cleaned vials [10 mL, 23 mm (diameter) × 46 mm (height), Chromoptic] spiked with nitrogen and then stored at −80 °C until analysis. 

Aliquots from one woman’s milk were used for extraction by the six methods investigated in this study. Discriminatory testing of the method that combined several SPME fibers and a Tenax trap in a dynamic extraction procedure (D-HS-MultiSPME/Tenax) was evaluated on the milk samples from the four remaining participants. Each analysis was performed in triplicate.

### 3.3. Extraction Headspace Methods

#### 3.3.1. Static Headspace Solid Phase Microextraction (HS-SPME-Car/PDMS) 

Volatile compounds from milk samples were extracted by static headspace solid-phase microextraction using Car/PDMS fiber (StableFlex, 10 mm length, 85 μm film thickness; Supelco, Bellefonte, PA, USA). The fiber was placed into the headspace of the vial for 1 h at 30 °C with stirring (400 rpm). Extraction was processed without heat incubation to preserve the extraction of compounds that could be degraded in a short-term period.

#### 3.3.2. Dynamic Headspace Extraction with Tenax or Trilayer Sorbent (D-HS-Tenax and D-HS-trilayer)

The milk sample was bubbled with a 100 mL·min^−1^ nitrogen flow for 1 h at 30 °C. Compounds were trapped onto sorbent(s) and placed into a glass tube (Markes International, Ltd., Llantrisant, UK) connected to the outlet of the sample vial and traversed by the nitrogen flow. Two cartridges were tested: one made of Tenax TA sorbent (200 mg) and a second made of a combination of Tenax TA, Carbograph 1TD and Carboxen 1003 sorbents with an equal bed length of each sorbent (summing 380 mg). The latest trilayer cartridge was chosen because of its graphitized carbon and molecular sieve content, which was expected to retain highly polar and volatile components. After extraction, the tubes were carried to the thermal desorption unit to be desorbed.

#### 3.3.3. Static Headspace with Multiple Solid Phase Microextraction Fibers (S-HS-MultiSPME)

In a first method application, ten SPME fibers (Car/PDMS, StableFlex, 10 mm length, 85 μm film thickness; Supelco) were simultaneously placed into the headspace of a milk sample for 1 h at 30 °C with stirring (400 rpm). After extraction, the fibers were placed into an empty glass tube and blocked with a glass wool plug and a tension spring to be further desorbed into the thermal desorption unit.

#### 3.3.4. Dynamic Headspace with Multiple Solid Phase Microextraction Fibers (D-HS-MultiSPME)

The 10 Car/PDMS SPME fibers were placed directly into an empty glass tube and blocked with a glass wool plug and tension spring in a second application. The milk sample was bubbled with a 100 mL·min^−1^ nitrogen flow for 1 h at 30 °C with the tube containing the fibers connected to the nitrogen flow outlet. Compounds were thus trapped onto the fibers placed into the glass tube. After extraction, the tube was similarly carried to the Thermal Desorption Unit.

#### 3.3.5. Dynamic Headspace with Multiple Solid Phase Microextraction Fibers and Tenax (D-HS-MultiSPME/Tenax)

In this last application, the 10 Car/PDMS SPME fibers were placed directly into a tube that contained 200 mg of Tenax TA. Fibers were separated from the Tenax by glass wool and were maintained in the tube by a glass wool plug and a tension spring. The milk sample was bubbled with a 100 mL·min^−1^ nitrogen flow for 1 hour at 30 °C so that the compounds were trapped in the tube containing the Tenax trap and SPME fibers and connected to the outlet of the nitrogen flow. After extraction, the tube was carried to the Thermal Desorption Unit.

### 3.4. Thermal Desorption and Gas Chromatography Analysis

#### 3.4.1. Desorption of Volatiles

For the conventional HS-SPME procedure, compounds were desorbed from the fiber directly in the injection port of the GC for 10 min at a temperature of 250 °C under splitless conditions.

For all other extraction procedures, whatever the glass tube cartridge, compounds were desorbed using the thermal desorption unit (TD, Unit 2 thermal desorption, Markes). A 50 mL·min^−1^ hydrogen flow was first applied at room temperature for 1 min to eliminate water, air and oxygen intrusion. Then, a two-stage desorption procedure was used following conditions previously described [43].

#### 3.4.2. GC/MS/FID Analyses

Analyses were carried out using a gas chromatograph (7890A GC System Agilent, Wilmington, DE, USA) equipped with a mass spectrometer (MS, 5975 inert MSD, Agilent) and a flame ionization detector (FID, Agilent). Conditions were based on previous work [43], adapting the temperature programming of the GC oven [50 °C (0 min), 5 °C.min^−1^ to 80 °C (0 min), 10 °C.min^−1^ to 200 °C (0 min), 20 °C.min^−1^ to 240 °C (15 min hold)]. 

### 3.5. Data Processing and Statistical Analyses

Peak areas were integrated using the MSD Chemstation software (Agilent). Compounds were identified by comparison of their mass spectra with those of reference databases (Wiley version 11, National Institute of Standards and Technology—NIST version 17 and internal laboratory database). Authentic compounds, when available, were also injected under the same conditions. Retention times and mass spectra thus obtained were compared with those of compounds detected. Linear retention indices (LRIs) of compounds were calculated using n-alkane injections (C5 to C26) and compared with values from the literature. One-way analyses of variance (ANOVA) and least significant difference (LSD) multiple comparison tests (95% confidence level) were performed on FID peak areas obtained with the different extraction methods. Coefficients of variation (CVs in %) were determined to compare the repeatability of each extraction method. CVs were calculated as the ratio of the standard deviation to the mean of the FID peak area values obtained (*n* = 3) for each compound and each extraction method. Values are expressed as a percentage. 

The evaluation of the discriminative ability of the D-HS-MultiSPME/Tenax method was displayed through the analysis of four individual milk samples. A principal component analysis (PCA) was applied to the FID peak areas of compounds (active variables) obtained from the analysis of individual milk samples collected from four women. In this analysis, the sum of the peak area of each chemical class was added as illustrative variables. XLSTAT software (version 011.2.08, Addinsoft, New York, NY, USA) was used to perform statistical analyses.

## 4. Conclusions

The present study aimed to perfect an extraction method of volatiles adaptable to small samples, valid in terms of repeatability, discriminative potency, and feasibility with minimal handling of the highly unstable human milk matrix. As the ultimate application of this extraction method was to identify molecular vectors bearing olfactory effect on newborn infants, the extraction procedure focused on the headspace developing over standing milk. Several extraction methods were designed after published procedures and the currently available sorbent polymers. Comparing the volatile profiles obtained by conventional HS-SPME-Car/PDMS and D-HS-Tenax extraction methods showed that D-HS-Tenax enables better detection and identification of a greater number of compounds in human milk headspace than HS-SPME-Car/PDMS. This result is primarily explained by the greater amount of sorbent used in the D-HS cartridge. However, the HS-SPME-Car/PDMS extraction appears more efficient in recovering short-chain fatty acids, commonly found in human milk (and amniotic fluid) and which are olfactorily active on newborn infants [56,64,65,66]. 

The combination of the powdered polar sorbent Carboxen with apolar Tenax through the D-HS-trilayer method led to the ice-plug formation in the desorption system and was thus not workable. Therefore, considering the strong extraction potency of the D-HS polymer and the large range of polar compounds recovered by HS-SPME-Car/PDMS, innovative procedural variations of the latter headspace methods were assessed to analyze volatile compounds from human milk. The first variation (S-HS-MultiSPME), consisting of 10 Car/PDMS SPME fibers in a static extraction procedure, was conceived to increase the sensitivity of the HS-SPME. It led to a 6-fold amplification of the global signal compared to the use of a single fiber. The second variation (D-HS-MultiSPME), combining the same 10 SPME fibers but in a dynamic extraction procedure, led to a lower increase of the GC signal (4-fold increase) compared to S-HS-MultiSPME. However, it better-recovered aldehydes, terpenes and alcohols relative to static extraction, resulting in extract profiles that resembled those obtained from conventional D-HS-Tenax extraction. Finally, considering the complementary sorbent potential of Car/PDMS fibers and the Tenax in terms of molecular polarity, a last procedural variation combined 10 Car/PDMS fibers with the Tenax sorbent in a unique dynamic extraction step (D-HS-MultiSPME/Tenax). This method widened the range of polarity of compounds recovered from human milk. Among the six extraction procedures tested in this study, it allowed the detection of compounds from the largest range of polarities and the highest concentrative yield with adequate repeatability. This association may constitute a suitable alternative to currently unavailable commercial solutions. 

D-HS-MultiSPME/Tenax method was demonstrated to be suitable for discriminating individual 5-mL human milk samples. Moreover, it identified more than 60 compounds in human milk samples belonging to the main chemical families reported in the literature: i.e., carboxylic acids, esters, alcohols, aldehydes, terpenes and ketones. In addition, this method recovered several compositionally minor compounds such as furans, lactones, pyrrole and pyrazine from human milk. Among them, pyrazine and the 2-methylpyrrole were, to our knowledge, identified for the first time in human milk. 

Finally, the D-HS-MultiSPME/Tenax extraction method constitutes a valuable tool to push forward the chemical exploration of the effluvium of human milk and colostrum. This new method can now be put into systematic use to delve deeper into the understanding of the compound(s) that convey the strongest informative value to newborn infants and are most active in eliciting their attention, attraction and appetence (e.g., [3,67]). More broadly, this extraction procedure can also be applied to analyze the volatilome emanating from other human biological fluids, even when these are available in very low amounts.

## Figures and Tables

**Figure 1 molecules-27-05299-f001:**
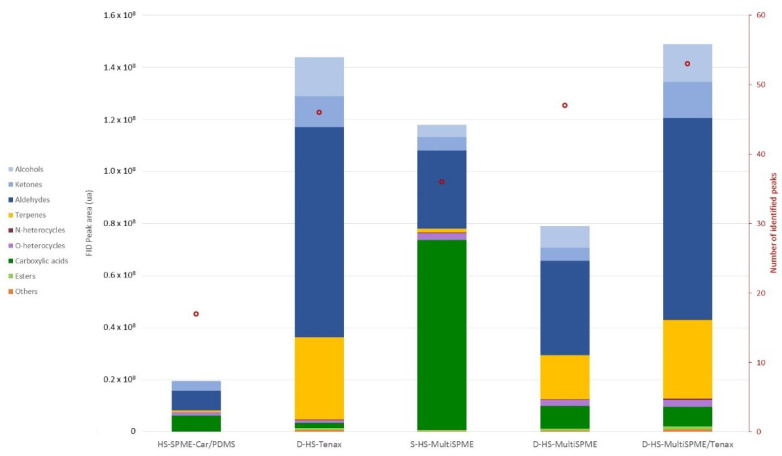
Total FID peak areas, number of peaks corresponding to compounds, and their belonging to different chemical classes obtained from human milk using the different headspace extraction methods.

**Figure 2 molecules-27-05299-f002:**
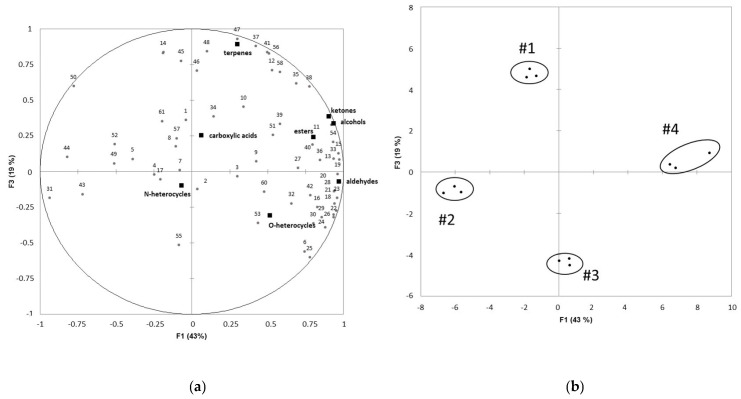
Principal component analysis performed on the peak areas of the volatile compounds detected in the milk samples of four individual donors. (**a**) Loading plot for volatile compounds (**b**) Scores plot for human milk samples. 1: acetic acid (LRI: 1460), 2: methanoic acid (LRI: 1534), 3: propanoic acid (LRI:1547), 4: butanoic acid (LRI: 1637), 5: pentanoic acid (LRI: 1746), 6: hexanoic acid (LRI: 1862), 7: heptanoic acid (LRI: 1968), 8: octanoic acid (LRI: 2076), 9: nonanoic acid (LRI: 2172), 10: decanoic acid (LRI: 2378), 11: 2-methyl-1-propanol (LRI: 1077), 12: 1-butanol (LRI: 1140), 13: (*E*)-1-penten-3-ol (LRI: 1145), 14: 3-methyl-1-butanol (LRI: 1205), 15: 1-pentanol (LRI: 1250), 16: 1-octen-3-ol (LRI: 1445), 17: 2-ethyl-1-hexanol (LRI: 1488), 18: pentanal (LRI: 980), 19: hexanal (LRI: 1088), 20: heptanal (LRI: 1190), 21: (*E*)-2-hexenal (LRI: 1233), 22: octanal (LRI: 1295), 23: (*E*)-2-heptenal (LRI: 1346), 24: nonanal (LRI: 1409), 25: (*E*)-2-octenal (LRI: 1445), 26: (*E*,*E*)-2,4-heptadienal (LRI: 1484), 27: decanal (LRI: 1498), 28: (*E*,*Z*)-2,4-heptadienal (LRI: 1521), 29: (*E*)-2-nonenal (LRI: 1563), 30: benzaldehyde (LRI: 1563), 31: 2-propanone (LRI: 793), 32: 2-butanone (LRI: 904), 33: 2,3-butanedione (LRI: 975), 34: 3-octanone (LRI: 1264), 35: 6-methyl-5-hepten-2-one (LRI: 1351), 36: 2-nonanone (LRI: 1397), 37: 3-octen-2-one (LRI: 1423), 38: 2-undecanone (LRI: 1611), 39: acetophenone (LRI: 1702), 40: ethyl acetate (LRI: 878), 41: methyl decanoate (LRI: 1601), 42: α-pinene (LRI: 1023), 43: camphene (LRI: 1071), 44: β-pinene (LRI: 1114), 45: 3-carene (LRI: 1156), 46: β-myrcene (LRI: 1163), 47: dl-limonene (LRI: 1210), 48: 1,8-cineole (LRI: 1220), 49: γ-terpinene (LRI: 1259), 50: α-terpinolene (LRI: 1295), 51: (*E*)-geranylacetone (LRI: 1885), 52: pyrazine (LRI 1226), 53: 2-methylpyrrole (LRI: 1575), 54: 2-pentylfurane (LRI 1236), 55: furfural (LRI: 1486), 56: 5-methyl-2-furfural (LRI: 1601), 57: toluene (LRI: 1041), 58: styrene (LRI: 1275), 59: Butyrolactone (LRI: 1685), 60: phenol (LRI: 2049).

**Table 1 molecules-27-05299-t001:** Volatile compounds identified in human milk sample using five headspace extraction methods: experimental linear retention index (LRIexperimental); identified compounds; their CAS number; major ions (*m*/*z*); LRI obtained from standard injection; LRI from the literature; mean of peak areas for compounds and their coefficient of variation (CV) obtained from the HS-SPME-Car/PDMS, D-HS-Tenax, S-HS-MultiSPME, D-HS-MultiSPME, D-HS-MultiSPME/Tenax methods; ANOVA significance and references for identification of compounds from literature.

LRI_experimental_	Compound	CAS	Major Ions (*m*/*z*)	LRI_STD_	LRI _Literature_	HS-SPME-Car/PDMS				D-HS-Tenax				S-HS-MultiSPME				D-HS-MultiSPME				D-HS-MultiSPME/Tenax				ANOVA Significance	Literature References
		Number				Mean Peak Area (×10^5^)			CV	Mean Peak Area (×10^5^)			CV	Mean Peak Area (×10^5^)			CV	Mean Peak Area (×10^5^)			CV	Mean Peak Area (×10^5^)			CV		
**Acids**																											
1460	Acetic acid	64-19-7	45, 60		1434–1457	9.89	^b^	±	>50%	10.6	^b^	±	4%	115	^a^	±	15%	1.57	^b^	±	32%	11.9	^b^	±	8%	***	[23,25,26,27,28]
1547	**Propanoic acid**	79-09-4	45, 57, 74	1540	1498–1564	-				-				9.56		±	45%	6.60		±	47%	13.0		±	35%	ns	-
1637	**Butanoic acid**	107-92-6	60, 73	1637	1619–1628	3.96	^c^	±	47%	-				52.6	^a^	±	8%	14.5	^b^	±	6%	13.1	^b^	±	2%	***	[25,26,27]
1746	**Pentanoic acid**	109-52-4	60, 73, 41	1746	1698–1720	-				-				5.46		±	41%	3.90		±	58%	3.00		±	23%	ns	[26,27,30]
1862	**Hexanoic acid**	142-62-1	60, 73, 87	1853	1810–1861	9.72	^b^	±	24%	-				91.9	^a^	±	6%	6.02	^b^	±	56%	3.94	^b^	±	11%	***	[25,31]
1968	**Heptanoic acid**	111-14-8	60, 87	1958	1915–1997	-				-				3.98		±	24%	2.92		±	45%	3.53		±	60%	ns	[25]
2076	**Octanoic acid**	124-07-2	43, 55, 60, 73, 85	2062	2030–2089	27.1	^b^	±	37%	4.48	^c^	±	49%	255	^a^	±	11%	14.1	^c^	±	29%	7.33	^c^	±	21%	***	[20,25,31]
2172	**Nonanoic acid**	112-05-0	43, 55, 60, 73	2266	2185–2277	-				-				14.0	^a^	±	22%	6.68	^ab^	±	52%	3.32	^b^	±	19%	*	[25]
2378	**Decanoic acid**	334-48-5	43, 55, 60, 74	>2200	2294–2316	11.8	^c^	±	29%	5.84	^c^	±	63%	181	^a^	±	20%	30.3	^bc^	±	48%	15.6	^c^	±	49%	***	[20,23,25,26,28,30,31]
	**Total**					**62.5**	** ^cd^ **			**20.9**	** ^d^ **			**728**	** ^a^ **			**86.6**	** ^b^ **			**74.7**	** ^c^ **			***	
**Alcohols**																											
1077	**2-methyl-1-propanol**	78-83-1	74	1080	1086–1114	-				31.5	^a^	±	2%	11.7	^b^	±	7%	14.0	^b^	±	8%	30.8	^a^	±	5%	***	-
1140	**1-butanol**	71-36-3	41, 56	1137	1143	-				64.0	^a^	±	10%	18.5	^c^	±	15%	41.0	^b^	±	8%	59.7	^a^	±	17%	***	[27,28]
1154	(*E*)*-1-penten-3-ol*		56			-				9.09	^a^	±	6%	1.65	^c^	±	30%	5.65	^b^	±	12%	9.46	^a^	±	4%	***	[27,30]
1205	**3-methyl-1-butanol**	123-51-3	70, 55, 41	1202	1180–1237	-				0.23	^b^	±	9%	-				-				2.42	^a^	±	24%	***	-
1250	**1-pentanol**	71-41-0	42, 55, 70	1245	1213–1261	1.61	^e^	±	22%	36.8	^b^	±	8%	7.51	^d^	±	7%	19.6	^c^	±	4%	40.3	^a^	±	6%	***	[27,28]
1488	2-ethyl-1-hexanol	104-76-7	41, 57, 70, 83, 98		1484–1492	-				7.35		±	20%	7.38		±	19%	4.32		±	27%	3.73		±	3%	ns	[25,27]
	**Total**					**1.61**	** ^d^ **			**149**	** ^a^ **			**46.7**	^c^			**85.6**	** ^b^ **			**146**	** ^a^ **			***	
**Aldehydes**																											
980	**Pentanal**	110-62-3	44, 58, 71, 86	980	973–988	4.8	^d^	±	23%	44.1	^a^	±	4%	23.1	^b^	±	10%	19.3	^c^	±	11%	46.3	^a^	±	2%	***	[25,27,28]
1088	**Hexanal**	66-25-1	44, 56	1085	1067–1097	70.4	^c^	±	14%	599	^a^	±	2%	222	^b^	±	7%	257	^b^	±	9%	585	^a^	±	5%	***	[20,23,25,26,27,28,29,30]
1190	**Heptanal**	111-71-7	55, 70, 81, 96	1191	1184–1198	-				26.7	^a^	±	6%	7.13	^c^	±	21%	14.2	^b^	±	27%	24.6	^a^	±	11%	***	[20,25,27,28,29]
1233	**(E)-2-hexenal**	6728-26-3	41, 55, 69, 83, 98	1231	1200–1230	-				9.80	^a^	±	6%	2.69	^d^	±	15%	4.16	^c^	±	5%	8.15	^b^	±	5%	***	[25,27,30]
1295	octanal	124-13-0	67, 84		1275–1306	-				3.36		±	7%	2.98		±	19%	2.58		±	15%	2.98		±	21%	ns	[23,25,26,27,28,30]
1346	(*E*)-2-heptenal	18829-55-5	41, 55, 70, 83		1313–1350	-				14.1	^a^	±	6%	4.39	^c^	±	8%	10.6	^b^	±	8%	14.9	^a^	±	17%	***	[20,25]
1409	Nonanal	124-19-6	57, 41, 70,82		1376–1416	-				58.1	^a^	±	11%	13.2	^d^	±	29%	24.1	^c^	±	24%	39.9	^b^	±	5%	***	[25,26,27,28,30]
1445	**(E)-2-octenal**	2548-87-0	41, 55, 70, 83, 97	1449	1423	-				10.6	^a^	±	4%	-				1.58	^b^	±	12%	11.9	^a^	±	8%	***	[25,26,27,30]
1484	(*E*,*E*)-2,4-heptadienal	4313-03-5	81, 110		1451–1497	-				7.52	^a^	±	2%	-				4.29	^b^	±	13%	7.46	^a^	±	3%	***	[25,27]
1498	Decanal	112-31-2	43, 57, 70, 82, 95, 112		1447–1500	-				9.06		±	12%	8.06		±	10%	6.38		±	28%	9.23		±	8%	ns	[25,26,27,28,30]
1521	(*E*,*Z*)-2,4-heptadienal	4313-02-4	81, 95		1516–1569	-				3.05	^a^	±	11%	-				2.05	^b^	±	9%	3.18	^a^	±	21%	***	[25]
1563	(*E*)-2-nonenal	18829-56-6	55, 70		1519–1569	-				9.84	^a^	±	5%	1.30	^c^	±	2%	4.30	^b^	±	28%	9.42	^a^	±	8%	***	[25,26,27,30]
1563	Benzaldehyde	100-52-7	51, 77, 105, 106		1508–1555	0.56	^b^	±	9%	12.1	^a^	±	3%	14.8	^a^	±	22%	10.1	^a^	±	33%	12.3	^a^	±	17%	**	[25,27,28]
	**Total**					**75.8**	** ^d^ **			**807**	** ^a^ **			**300**	** ^c^ **			**361**	** ^b^ **			**775**	** ^a^ **			***	
**Carbonyls/ketones**																											
793	2-propanone	67-64-1	43, 58		814–845	26.9		±	32%	-				-				-				-					[25,27,28]
904	**2-butanone**	78-93-3	43, 72, 57	888	917–949	2.13	^b^	±	31%	4.86	^a^	±	28%	-				-				2.93	^b^	±	9%	***	[25,27]
975	**2,3-butanedione**	431-03-8	43, 86	966	962–1020	9.59	^c^	±	23%	88.3	^a^	±	4%	46.3	^b^	±	10%	38.6	^b^	±	11%	92.7	^a^	±	2%	***	[25,26]
1264	3-octanone	106-68-3	43, 57, 71, 99, 128		1240–1272	-				1.23	^c^	±	14%	1.35	^c^	±	30%	3.18	^a^	±	23%	2.28	^b^	±	3%	***	[25,27]
1351	**6-methyl-5-hepten-2-one**	110-93-0	43, 55, 69, 83, 93	1345	1313–1367	-				10.1	^b^	±	10%	1.23	^c^	±	16%	1.97	^c^	±	13%	22.6	^a^	±	21%	***	[25]
1397	**2-nonanone**	821-55-6	58, 43	1396	1393	-				3.14	^a^	±	8%	1.03	^c^	±	29%	2.26	^b^	±	30%	3.29	^a^	±	11%	***	[25,28]
1423	**3-octen-2-one**	1669-44-9	111, 55, 43, 126	1421	1363–1429	-				3.87		±	15%	-				-				3.73		±	27%	ns	[25,27]
1525	3,5-octadien-2-one	38284-27-4	81, 95		1516–1569	-				4.64	^a^	±	7%	-				1.38	^b^	±	27%	5.65	^a^	±	29%	***	[25]
1611	2-undecanone	112-12-9	43, 58, 71		1635	-				3.26	^b^	±	21%	1.07	^d^	±	14%	2.59	^c^	±	13%	4.95	^a^	±	3%	***	[25]
	**Total**					**36.6**	** ^c^ **			**119**	** ^b^ **			**51**	** ^c^ **			**50.0**	** ^c^ **			**138**	** ^a^ **			***	
**Esters**																											
878	**Ethyl acetate**	141-78-6	43, 61, 70	867	884–908	-				3.90	^a^	±	12%	-								2.95	^b^	±	12%	***	[25,28]
1601	Methyl decanoate	110-42-9	74, 87		1597–1614	-				1.04	^b^	±	15%	-				1.10	^b^	±	27%	2.63	^a^	±	23%	***	[28]
1644	Ethyl decanoate	110-38-3	88, 101		1595–1660	-				1.88	^b^	±	10%	5.37	^a^	±	22%	6.80	^a^	±	15%	6.49	^a^	±	25%	***	[28]
	**Total**					-				**6.82**	** ^bc^ **			**5.37**	** ^c^ **			**7.90**	** ^b^ **			**12.07**	** ^a^ **			***	
**Terpenes**																											
1114	**β-pinene**	127-91-3	93, 69, 77, 106, 121	1114	1094–1124	-				3.34	^a^	±	5%	-				0.96	^b^	±	22%	3.12	^a^	±	16%	***	
1156	**3-carene**	13466-78-9		1156	1118–1161	-				4.54	^a^	±	6%	-				0.29	^b^	±	14%	4.06	^a^	±	4%	***	[27]
1163	**β-myrcene**	123-35-3	93, 69, 41	1162	1155–1166	-				13.1	^a^	±	26%	-				4.48	^b^	±	9%	13.8	^a^	±	11%	***	[27]
1174	β-phellandrene	555-10-2	41, 69, 93		1163–1212	-				1.72		±	6%	-				1.22		±	22%	1.50		±	23%	ns	-
1210	DL-limonene	138-86-3	68, 93, 79		1185–1200	5.62	^e^	±	36%	254	^a^	±	2%	13.0	^d^	±	19%	148	^c^	±	3%	243	^b^	±	2%	***	[20,25,28]
1220	**1,8-cineole**	470-82-6	77, 93, 121, 136	1218	1199–1253	0.65	^e^	±	25%	27.7	^a^	±	3%	1.98	^d^	±	13%	13.1	^c^	±	7%	25.9	^b^	±	1%	***	[28]
1259	**γ-terpinene**	99-85-4	43, 77, 93, 121, 138	1252	1224–1266	-				5.10	^a^	±	7%	-				1.58	^c^	±	30%	3.42	^b^	±	10%	***	[27]
1295	α-terpinolene	586-62-9	93, 121, 136		1278–1288	-				1.66	^a^	±	7%	-				1.29	^b^	±	15%	1.49	^ab^	±	21%	***	[27]
1885	**(E)-geranylacetone**	3796-70-1	43, 69	1875	1840–1865	-				3.75		±	62%	-				-				6.51		±	74%	ns	[25]
	**Total**					**6.27**	** ^e^ **			**315**	** ^a^ **			**15.0**	** ^d^ **			**171**	** ^c^ **			**303**	^b^			***	
**N-Heterocycles**																											
1226	Pyrazine	290-37-9	80, 53		1209–1257	-				3.38		±	32%	2.22		±	33%	2.45		±	29%	2.94		±	21%	ns	-
1575	2-methylpyrrole	636-41-9			1551–1570	-				-				-				-				1.13		±	14%		-
	**Total**					-				**3.38**	** ^b^ **			**2.22**	** ^b^ **			**2.45**	** ^b^ **			**4.07**	** ^a^ **			*	
**O-heterocycles**																											
1486	Furfural	98-01-1	39, 96		1436–1494	10.5		±	>100%	7.35		±	20%	7.08		±	12%	3.32		±	11%	3.73		±	3%	ns	[20,27,28]
1685	**γ-butyrolactone**	96-48-0	42, 56, 86	1682	1610–1640	2.66		±	29%	2.34		±	24%	19.4		±	28%	19.2		±	33%	23.5		±	18%	ns	-
	**Total**					**13.2**				**9.7**				**28.1**				**22.5**				**27.2**				ns	
**Other**																											
1275	Styrene	100-42-5	104, 78, 51		1247–1260	0.43	^e^	±	19%	7.53	^b^	±	9%	3.43	^d^	±	15%	5.59	^c^	±	3%	9.39	^a^	±	14%	***	[12]
	**Total**					**199**	** ^e^ **			**1440**	** ^b^ **			**1179**	** ^c^ **			**803**	** ^d^ **			**1490**	** ^a^ **			***	

Compounds in bold are those identified by the three means of identification (MS, LRI and standard). Compounds identified by one or two mean(s) of identification are considered putatively identified. Asterisks indicate differences between peak area values obtained from the different extraction methods according to a one-way analysis of variance (* *p*  ≤  0.05; ** *p*  ≤  0.01; ****p*   ≤  0.001). NS indicates no differences between peak area values. Different letters (a, b, c, d and e) between columns indicate significant differences between extraction methods according to the least significant difference test (*p*  ≤  0.05). Some compounds identified in this study had already been detected in human milk according to at least references quoted [12,20,23,25,26,27,28,29,30,31].

## Data Availability

Not applicable.

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
