# Peer review of "Comparative Investigation of Conventional and Innovative Headspace Extraction Methods to Explore the Volatile Content of Human Milk"

_molecules, 2022, doi:10.3390/molecules27165299_

Round 1

Reviewer 1 Report

This paper is potentially interesting but there are some issues that should be carefully addressed by authors before making the paper suitable for publication in the Molecules.

Abstract is not in line with the rest of the manuscript: Authors should mention six methods investigated in their study.

Line 53: Please add examples of contaminants.

Lines 54-56: Please add examples for ‘some other compounds’

Line 165: Why did you use both the FID area and mass spectrometry responses for the interpretation of results. Please explain in the text.

Figure 1: Why did you use FID peak area? What about MS peak area?

Titles of the sections 2.1.5. and 2.1.6. are the same.

Line 417: How did you calculate CV? Please explain.

Did you perform any analytical validation (precision, LOD, LOQ, linear range)?

Material and Methods should be rewritten in order to decrease text similarity with Ref 43. In the case when methodology is the same as previously published (e.g. temperature program in GC), please add reference.

Reviewer 2 Report

Thanks to submit the manuscript "Comparative investigation of conventional and innovative

 headspace extraction methods to explore the volatile content of

 human milk" to Molecules. In this work, several extraction methods were used to evaluate the aromatic profile of human milk. The research was well conducted and the manuscript is well written. However, I have some considerations.

Line#36-68: in general the introduction is very long and therefore the reading is extremely. My suggestion is to summarize these lines in just one paragraph since it is only an introduction to reach the objective of the work, which is the evaluation of aroma research methodologymethodology. 

Figure 1: To improve the visualization of the figure, please use solid colors. 

Line#210: Are the authors of this work indicating that the presence of furfural in this work may be related to storage? Have these adverse conditions not been minimized?

Line#219: This statement needs to be referenced if it is kept. Some discussion points in this work leave doubts about the quality of the samples and this is one of them.

R&D:

 From what was described in the introduction, I expected the discussion to be focused on explaining why the volatile compounds that were found in milk were there and their link to the infant and its need for breastfeed. However, the discussion was focused on the conservation of the product or its comparison with other milks. It would be more interesting if the authors left the discussion more focused on the emotional or nutritional pattern of the Infant.

Round 2

Reviewer 1 Report

The manuscript is improved.